

**The possibility of rainfall estimation using R(Z,$Z_{DR}$,$K_{DP}$,$A_H$) :**
**A case study of heavy rainfall on 25 August 2014 in Korea**
**C.-H. You[1], M.-Y. Kang[2] and D.-I. Lee[1,2]**
[1]{Atmospheric Environmental Research Institute, Pukyong National University, Busan,
South Korea}
[2]{Department of Environmental Atmospheric Sciences, Pukyong National University,
Busan, South Korea}
Correspondence to: D.-I. Lee (leedi@pknu.ac.kr)
**Abstract**
To improve the accuracy of polarimetric rainfall relations for heavy rainfall, an extreme
rainfall case was analysed and some methods were examined. The observed differential
reflectivity ($Z_{DR}$) quality check was theoretically investigated using the relation between the
standard deviation of differential reflectivity and cross correlation, and the light rain method
for $Z_{DR}$ bias was also applied to the rainfall estimation. The best performance for this heavy
rainfall case was obtained when the moving average of $Z_{DR}$ over a window size of 9 gates was
applied to the rainfall estimation using horizontal reflectivity ($Z_H$) and $Z_{DR}$ and to the
calculation of $Z_H$ bias. The differential reflectivity calculated by disdrometer data may be an
alternative to the vertical pointing scan for calculating $Z_{DR}$ bias. The accuracy of the
combined rainfall relation, R(Z,$Z_{DR}$,$K_{DP}$,$A_H$) was relatively insensitive to $Z_{DR}$ and $Z_H$ biases in
both observations and simulations.
**1  Introduction**
Weather radar is a very useful remote sensing instrument for estimating rainfall amount due to
its high spatial and temporal resolution compared with other instruments. Calculations of
radar rainfall are based on the relationship between reflectivity ($Z$) and rain rate ($R$) known as
the *Z–R* relation (hereafter *R(Z)*). Experimentally measured drop size distributions (DSDs)
have been extensively used to obtain both radar reflectivity and rain rate (Compos and





Zawadzki, 2000). It can be shown that there is no unique global R(Z) relation because DSDs
can vary from storm to storm and even within the storm itself (You et al., 2010). There have
been a few studies on the calculation of the R(Z) relationship using disdrometer data with
rainfall types and rain gage adjusted rainfall amount for operational Doppler weather radars in
Korea (Jang et al., 2004; You et al., 2004; Suk et al., 2005).
Radar rainfall estimation may be contaminated by uncertainties such as hardware calibration,
partial beam filling, rain attenuation, bright band, and non-weather echoes (Wilson and
Brandes, 1979; Austin, 1989). To mitigate these problems, particle identification algorithms
have been developed using polarimetric parameters for improving data quality control and
rainfall estimates by discriminating non-meteorological artefacts such as anomalous
propagation, birds, insects, second trip echo, and melting layer detection (Ryzhkov and Zrnic,
1998; Vivekanandan et al., 1999; Giangrande et al., 2008). The improvement of radar rainfall
accuracy is a major reason for using polarimetric radar (Ryzhkov and Zrnic, 1996; May et al.,
1999; Bringi and Chandrasekar, 2001). Ryzhkov et al. (2005a) developed a rainfall algorithm
using polarimetric radar for the prototype WSR-88D (Weather Surveillance Radar-88 Doppler)
system using different drop shape assumptions. Ciffelli et al. (2011) compared two rainfall
algorithms, CSU-HIDRO (Colorado State University-Hydrometeor IDentification of Rainfall)
and JPOLE (Joint Polarization Experiment)-like, in the high plains environment. Ryzhkov et
al. (2014) recently investigated the potential use of specific attenuation ($A_H$) for rainfall
estimation with X-band and S-band radar and found that the R($A_H$) method yields robust
estimates of rain rates even at S band where attenuation is very small.
As a result of these theoretical and other experimental studies, many countries are replacing or
modifying their radars and using polarimetric radar operationally. There are three major
agencies that operate radars to monitor and forecast severe weather and flash flooding
operationally in Korea: the Ministry of National Defense (MND), the Ministry of Land,
Infrastructure and Transportation (MoLIT), and the Korea Meteorological Administration
(KMA), with the MoLIT the first to install polarimetric radars in Korea. The KMA installed
an S-band polarimetric radar in the far northwest of Korea in 2014. For successful operational
implementation of these radars, considerable research on rainfall estimation, hydrometeor
classification, and DSD retrieval is required. However, there have been few studies on these
polarimetric related issues other than the derivation of relationships using long period
disdrometer data and the assessment of each relation after applying a very simple quality





control for differential phase shift (You et al., 2014). You et al. (2014) found that the accuracy
of rainfall estimation using horizontal reflectivity ($Z_H$) and differential reflectivity ($Z_{DR}$)
obtained by DSDs in the Busan area in Korea was better than that obtained with relations
calculated by DSDs measured in Oklahoma in the US. A quality control algorithm and
unfolding of differential phase shift ($\Phi_{DP}$) for calculating specific differential phase ($K_{DP}$)
were applied to the rainfall estimation (You et al., 2014). Recently, You et al. (2015a)
proposed a relation combining many polarimetric variables of the form R(Z,$Z_{DR}$,$K_{DP}$,$A_H$) as a
candidate for an optimum rainfall relation for S-band polarimetric data in Korea; this would
allow a single relation to be used for different hydrometeor regimes in the absence of a stable
hydrometeor classification algorithm. However, there are still issues to be resolved in
improving $Z_{DR}$ data quality and the robustness of R(Z,$Z_{DR}$,$K_{DP}$,$A_H$) for the heavy rainfall case
where error propagation from each polarimetric variable can occur.
This paper discusses how to improve the accuracy of rainfall estimation using moving
averaged differential reflectivity and examines the robustness of the R(Z,$Z_{DR}$,$K_{DP}$,$A_H$) relation
for a heavy rainfall case in Korea. Sect. 2 describes the rain gage, DSD and radar dataset,
together with the calculation of polarimetric variables from DSDs and the validation methods.
Sect. 3 provides $Z_H$ and $Z_{DR}$ bias correction, an examination of $Z_{DR}$ data quality, and the
statistical results of rainfall estimation using observed and moving-average $Z_{DR}$. Sect. 4
contains a discussion of a possible method for improving R(Z,$Z_{DR}$) accuracy and the
robustness of the R(Z,$Z_{DR}$,$K_{DP}$,$A_H$) relation. Finally, we provide some conclusions in Sect. 5.

## 2   Data and methodology

### 2.1   Gage, disdrometer and radar dataset

The rainfall data from rain gages operated by the KMA were used to evaluate the accuracy of
radar rainfall. Rain gages located within the radar coverage area at distances from 5 to 95 km
of the radar are included in the analysis. Fig. 1 shows the location of all instruments used in
this study. The circle is the radar coverage, the solid rectangle is the centre of the Bislsan
radar, the plus signs show the rain gages within the radar coverage and the open rectangle is
the location of a PARSIVEL (PARticle Size VELocity) and POSS (Precipitation Occurrence




Sensor System; detailed specifications are provided by Sheppard, 1990) disdrometer installed
~82 km away from the radar.
Relations for converting radar variables into rain rate are required because the radar does not
observe rainfall directly. To calculate these relations, disdrometer data that can measure the
DSDs are needed. One-min DSDs obtained by the POSS from 2001 to 2004 were used. To
improve the accuracy of $Z_{DR}$, DSDs observed by PARSIVEL on 25 August 2014 were used
because POSS data were not available at that time. The PARSIVEL disdrometer is a laser-
optic system that measures 32 channels from 0.062 to 24.5 mm (detailed specifications are
given by Loffler-Mang and Joss, 2000).
Unreliable data, defined as belonging to the following categories, were removed: 1-min rain
rate less than 0.1 mm h$^{-1}$; total number concentrations of all channels less than 10; drop
numbers counted only in the lower 10 channels (0.84 mm for POSS and 1.187 mm for
PARSIVEL); and drop numbers counted only in the lower 5 channels (0.54 mm for POSS and
0.562 mm for PARSIVEL) (You et al., 2015b).
Radar data were collected by the Bislsan polarimetric radar installed and operated by the
MoLIT in Korea since 2009. The transmitted peak power is 750 kW, beam width is 0.95°, and
frequency is 2.791 GHz. The polarimetric variables are estimated with a gate size of 0.125 km.
The scan strategy is composed of 6 elevation angles with 2.5-min update interval.
Polarimetric variables for 0.5° elevation angle were extracted from the volume data every 10
mins for this study.
**2.2  Calculation of polarimetric variables from DSDs**
Polarimetric variables were calculated using T-matrix scattering techniques derived by
Waterman (1971) and later developed further by Mishchenko et al. (1996). The following
raindrop shape assumptions are used for the calculation of variables from the DSDs:
$$\frac{b}{a} = 1.0048 + 0.500057\,D - 0.02628\,D^2 + 0.003682\,D^3 - 0.0001677\,D^4, \quad (1)$$
$$\frac{b}{a} = 1.012 - 0.01445\,D - 0.01028\,D^2, \quad\quad\quad\quad\quad (2)$$
where a, b and D are the major axis, minor axis, and equi-volume diameter of raindrop in
millimetres, respectively.





Eq. (1) is for the equilibrium axis ratio derived from the numerical model of Beard and Chung
(1987), which is in good agreement with the results from wind tunnel measurements. The
actual shapes of raindrops in turbulent flow are expected to be different from the equilibrium
shapes due to drop oscillations. Oscillating drops appear to be more spherical on average than
drops with equilibrium shapes as shown by Andsager et al. (1999) in laboratory studies. They
demonstrated that the shape of raindrops with diameter between 1.1 and 4.4 mm is better
explained by Eq. (2). You et al. (2015a) found that combining Eq. (1) for drops less than 1.1
mm and larger than 4.4 mm with Eq. (2) for the drop diameter between 1.1 and 4.4 mm as
proposed by Bringi et al. (2003) gave the best rainfall estimation compared with other drop
axis ratio assumptions in Korea, and we use this combined formulation in this study. Other
parameters in the T-matrix calculations include the temperature, which is assumed to be 20℃
in this study. The distribution of canting angles of raindrops is Gaussian with a mean of 0°
and a standard deviation of 7°, as determined recently by Huang et al. (2008).

## 2.3 Validation

The localized rainfall on 25 August 2014 was caused by a low pressure system that passed
through southern Korea. Fig. 2 shows the time series of hourly rainfall and accumulated
rainfall from the three gages, ID 255 (North Changwon site), ID 926 (Jinbook site), and ID
939 (Geumjeong-gu site) that recorded the highest rainfall within the radar coverage area. The
daily accumulated rainfall values were 243.5 mm, 269.0 mm, and 244.5 mm for these gages.
The time period analysed was from 0900 LT to 1600 LT because the rainfall was
concentrated in this period and radar data were available from 0900 LT.
The normalized error (NE), fractional root mean square error (RMSE), and correlation
coefficients (CC) of the rainfall relations and 121 gages were used to investigate the
performance of each rainfall relation:

$$NE = \frac{\frac{1}{N}\sum_{i=1}^{N}\left|R_{R,i} - R_{G,i}\right|}{\overline{R_G}}, \qquad (3)$$

$$RMSE = \left[\frac{1}{N}\sum_{i=1}^{N}(R_{R,i} - R_{G,i})^2\right]^{1/2}, \qquad (4)$$



$$CC = \frac{\sum_{i=1}^{N}(R_{R,i} - \overline{R_R})(R_{G,i} - \overline{R_G})}{\left[\sum_{i=1}^{N}(R_{R,i} - \overline{R_R})^2\right]^{1/2}\left[\sum_{i=1}^{N}(R_{G,i} - \overline{R_G})^2\right]^{1/2}}, \qquad (5)$$
where N is the number of radar rainfall ($R_R$) and gage rainfall ($R_G$) pairs, and $\overline{R_R}$ and $\overline{R_G}$ are
the average hourly rain rates from the radar and gage, respectively. These statistical variables
are calculated using hourly rainfall amounts derived from the radar and gage at the location of
the gage. The radar rainfall at the rain gage was obtained by averaging rainfall over a small
area (1 km × 1°) centered on each rain gage. The rainfall relations for calculating radar
rainfall were obtained from the simulated polarimetric variables generated from DSDs and are
summarized in Table 1.
**3  Results**
**3.1  Improvement of $Z_{DR}$ data quality**
$Z_{DR}$ is an important variable for hydrometeor classification and rainfall estimation. To check
the quality of the $Z_{DR}$ measurements, the radial profile of $Z_{DR}$ was investigated as shown in
Fig. 3. Fig. 3(a) shows the spatial distribution of $Z_{DR}$ at 0.5° elevation at 1401 LT on 25
August 2014. Fig. 3(b) shows the radial profile of observed $Z_{DR}$ (red line) and the standard
deviation of $Z_{DR}$ (black line) calculated using 9 gates along the line A–B shown in Fig. 3(a).
The average standard deviation of $Z_{DR}$ along the line was 0.615 dB. Fig. 3(c) shows the radial
profile of the cross correlation; the average cross correlation was 0.982.
To find the accuracy of the observed $Z_{DR}$ value, we use the theoretical relation between the
standard deviation of $Z_{DR}$ and the cross correlation following Bring and Chandrasekar (2003):
$$SD(Z_{DR}) = 10\log_{10}\left\{1 + \left[\frac{2}{N}(1 - |\rho_{co}|^2)\sum_{l=-(N-1)}^{N-1}(1 - \frac{|l|}{N})|\rho_{co}(l)|^2\right]^{1/2}\right\}, \qquad (6)$$
where SD($Z_{DR}$) is standard deviation of $Z_{DR}$, N is the number of samples and $\rho_{co}$ is the cross
correlation, given by




$\rho[n] = \exp(-\frac{8\pi^2 \sigma_v^{\ 2} n^2 T_s^{\ 2}}{\lambda^2})$,        (7)
where $\rho$ is the cross correlation, $\sigma_v$ is Doppler width, n is sample number, and $T_s$ is dwell
time.
For a better comparison we display the correlations in $L$ space, as proposed by Keat et al.

5    (2015)

$L = -10 \log_{10}(1 - \rho_{hv})$,        (8)
where, $\rho_{hv}$ is cross correlation. Fig. 4 shows the theoretical relation between the standard
deviation of $Z_{DR}$ and the cross correlation coefficient. Fig. 4(a) shows the results obtained
using the scan configuration of the Bislsan radar. The dwell time is 56 ms, number of samples
is 55, and the normalised Doppler width is 0.02. Fig. 4(a) suggests that for an accuracy of 0.1
dB in $Z_{DR}$ with 1 ms$^{-1}$ Doppler width, a value of L of over 3 ( $\rho_{hv}$ >0.999) is needed. Such
values cannot be measured with the antenna. In Fig. 4(b) the number of samples is 495, which
corresponds to 9 gates, 1.125 km in range; an accuracy of 0.2 dB in $Z_{DR}$ (the moving-average
$Z_{DR}$, hereafter m$Z_{DR}$) is achieved with 1 ms$^{-1}$ Doppler width and a value of L of 1.7 ( $\rho_{hv}$
>0.980).
Fig. 5 shows the results for $Z_{DR}$ measurements at 1401 LT on 25 August 2014. Fig. 5(a)
shows the spatial distribution of a moving average $Z_{DR}$ from 9 gates. Fig. 5(b) shows the
radial profile of the $Z_{DR}$ (red line) and its standard deviation (black line) calculated for 9 gates
along the line A–B shown in Fig. 5(a). The average standard deviation of $Z_{DR}$ along the ray
was 0.169 dB. Fig. 5(c) shows the radial profile of the cross correlation; the average cross
correlation was 0.985. Both the standard deviation of $Z_{DR}$ and the averaged $\rho_{hv}$ values are
very close to the theoretical values (standard deviation of $Z_{DR}$ is 0.160 and $\rho_{hv}$ is 0.987) as
shown in Fig. 4. Therefore, in the next Sect. a 9-gage moving average $Z_{DR}$ was used for
absolute $Z_H$ bias correction and rainfall estimation, and its effect on the accuracy of radar
rainfall estimation was examined.



### 3.2 Absolute bias correction of $Z_{DR}$ and $Z_H$
Before calculating radar rainfall, the $Z_H$ and $Z_{DR}$ must be corrected for system bias. Ryzhkov
et al. (2005) calculated the required accuracy for classifying light rain and dry snow to be 1
dB and 0.2 dB for $Z_H$ and $Z_{DR}$, respectively. The $Z_{DR}$ bias correction is important for the
absolute calibration of the radar using the self-consistency method. Gorgucci et al. (1999)
proposed a vertical pointing scan of light rain to take advantage of the nearly spherical shape
of the raindrops seen from below.
Ryzhkov et al. (2005b) used the elevation angle dependency of $Z_{DR}$ as an alternative
technique and concluded that the high variability of $Z_{DR}$ in rainfall means it is not possible to
achieve the required absolute calibration of 0.2 dB. They also proposed a method using the
structural characteristics of the melting layer in stratiform clouds and measured the dry
aggregated snow present above the melting layer, which gave a mean value of 0.2 dB at S
band and an accuracy of 0.1 to 0.2 dB.
Trabal et al. (2009) evaluated two different methods using the intrinsic properties of dry
aggregated snow present above the melting layer and measurements of light rain close to the
ground and found that a $Z_{DR}$ calibration accuracy of 0.2 dB or less was achieved for both
events analysed when both methods are compared.
The vertical pointing data were not available for the case considered here and the scan
strategy with six elevation angles does not detect the melting layer. Therefore, light rain
measurements close to the ground were used to calibrate the $Z_{DR}$ and $Z_H$ biases using the self-
consistency method in this study. Very light rain was defined by the thresholds 20 dBZ ≤ $Z_H$
≤ 28 dBZ as proposed by Marks et al. (2011). The $Z_H$ bias was determined following
Ryzhkov et al. (2005b).
The $Z_H$ biases calculated with the self-consistency method using observed $Z_{DR}$ and $mZ_{DR}$ are
−1.95 dB and −1.48 dB, respectively. The $Z_{DR}$ biases calculated by the very light rain method
using observed $Z_{DR}$ (0.26 dB) and $mZ_{DR}$ (0.3 dB), respectively.
### 3.3 Validation
To investigate the performance of R(Z,$Z_{DR}$) and R(Z,$Z_{DR}$,$K_{DP}$,$A_H$), which is related to the $Z_H$
and $Z_{DR}$ bias, NE, RMSE, and CC were calculated using hourly rainfall from each relation





and from the gages. For the comparison of rainfall amount, two different $Z_H$ and $Z_{DR}$ biases
were applied to observed variables as mentioned in Sect. 3.2. Each bias was calculated using
observed $Z_{DR}$ and $mZ_{DR}$.
Fig. 6 shows the scatter plot of 1 hour rainfall obtained using $R(Z,Z_{DR})$ and gage data. In Fig.
6 (a) the $Z_H$ bias was obtained from the observed $Z_{DR}$ bias and the $Z_{DR}$ biases calculated from
observed $Z_{DR}$ (blue full circles) and $mZ_{DR}$ (red full circles). The RMSE, NE, and CC of the
relation using $mZ_{DR}$ were as much as 8 mm h$^{-1}$, 0.1, and 0.18 better than those obtained using
observed $Z_{DR}$, respectively. In Fig. 6(b) the $Z_H$ bias is calculated from $mZ_{DR}$; the improved
performance using $mZ_{DR}$ is clear. The accuracy of the rainfall estimate using $Z_H$ bias obtained
by $mZ_{DR}$ is statistically more robust than that for the estimate based on observed $Z_{DR}$. The
RMSE, NE, and CC for the comparison of $R(Z,Z_{DR})$ rainfall obtained using different $Z_H$ and
$Z_{DR}$ biases are summarized in Table 2.
Fig. 7 shows the scatter plots when $R(Z,Z_{DR},K_{DP},A_H)$ is used for rainfall estimation. Fig. 7(a)
shows the radar rainfall calculated using the $Z_H$ bias obtained from the observed $Z_{DR}$ bias and
the $Z_{DR}$ biases obtained from observed $Z_{DR}$ (blue full circles) and $mZ_{DR}$ (red full circles). The
RMSE, NE, and CC from each relation were not very different; differences of RMSE, NE,
and CC in the two cases were 0.2 mm h$^{-1}$, 0.01, and 0, respectively. The statistics for the
comparison of radar rainfall obtained using different $Z_H$ and $Z_{DR}$ biases are summarized in
Table 3. These results show that $R(Z,Z_{DR},K_{DP},A_H)$ is less sensitive to $Z_H$ and $Z_{DR}$ error than
$R(Z,Z_{DR})$. This will be discussed further in Sect. 4.2 using simulated data.
**4    Discussion**
**4.1    Impact of disdrometer data on radar rainfall**
In the cases described in Sect. 3.3, the accuracy of the $R(Z,Z_{DR})$ relation was improved when
the moving-average $Z_{DR}$ (i.e., $mZ_{DR}$) was used to estimate rainfall. To improve the accuracy of
rainfall estimation using $R(Z,Z_{DR})$, we examined the impact of $Z_{DR}$ bias (as obtained from
disdrometer data) on the accuracy. The DSD data were quality controlled and polarimetric
variables were calculated by T-matrix simulation with the same configuration as in Sect. 2.
Before applying the DSDs to rainfall estimation, 10-min rainfall amounts obtained by DSDs
and gages were compared.





Fig. 8 shows the scatter plot of 10-min rainfall amount measured by PARSIVEL and the gage
located less than 100 m away from PARSIVEL. The daily accumulated rainfall amounts were
116.0 mm for the gage and 129.4 mm for PARSIVEL. The RMSE, NE, and CC were 0.52
mm, 0.26, and 0.99, respectively. For the comparison the $Z_{DR}$ of the radar was averaged over
3 km × 3° as shown in Fig. 9. The calculated $Z_{DR}$ biases were 0.26 dB for observed $Z_{DR}$ and
0.30 dB for $mZ_{DR}$. The $Z_H$ biases described in Sect. 3.3 were used.
Fig. 10 shows the scatter plots of 1-hour rainfall obtained by $R(Z,Z_{DR})$ and gages. The radar
rainfall was calculated after $Z_{DR}$ bias correction using the bias result in the comparison
between radar $Z_{DR}$ and PARSIVEL $Z_{DR}$. The $Z_{DR}$ biases were –0.05 dB for observed $Z_{DR}$ and
–0.07 dB for $mZ_{DR}$. In Fig. 10 (a) the $Z_H$ bias was obtained from the observed $Z_{DR}$ bias and
$Z_{DR}$ biases calculated from observed $Z_{DR}$ (blue full circle) and $mZ_{DR}$ (red full circle). The
radar rainfall using $mZ_{DR}$ was better than that using observed $Z_{DR}$ by as much as 5.5 mm h$^{-1}$
for RMSE and 0.36 for NE. In Fig. 10 (b) the $Z_H$ bias was calculated from $mZ_{DR}$; the
improved rainfall estimation using $mZ_{DR}$ is clear. This result shows the better scores
compared with the statistics shown in Fig. 6 that were obtained using $Z_{DR}$ biases extracted
from the radar $Z_{DR}$ only. When the observed $Z_{DR}$, which fluctuates considerably along the ray,
was applied to the rainfall estimation, the rainfall amount was much more variable with $Z_H$
bias values (blue full circle) than that with $mZ_{DR}$ (red circle) as shown in Fig. 10 (a) and (b).
According to these results, when moving average $Z_{DR}$ (i.e., $mZ_{DR}$) is used with the $Z_{DR}$ bias
measured by PARSIVEL, the accuracy of rainfall estimation was improved and was more
stable than that of other configurations using $R(Z,Z_{DR})$.
Fig. 11 shows the scatter plots for $R(Z,Z_{DR},K_{DP},A_H)$ and gages. The statistical scores were not
very different from $Z_H$ and $Z_{DR}$ biases. The differences of RMSE, NE, and CC between each
relation were 0.4 mm h$^{-1}$, 0.01, and 0, respectively. These results were summarized in Table 3.
**4.2  Simulation of $R(Z,Z_{DR},K_{DP},A_H)$ with error propagation from each variable**
With the relation using combined polarimetric variables, $R(Z,Z_{DR},K_{DP},A_H)$, error propagation
can affect the accuracy of radar rainfall estimation. To examine the contribution of errors
from each variable, simulated polarimetric variables such as Z, $Z_{DR}$, $K_{DP}$, $A_H$, were generated
with dimensions of 960 sizes of bins and 360 radials.
Fig. 12 shows the distribution function of the polarimetric variables generated assuming a
Gaussian distribution in each case. Fig. 12(a) shows the occurrence frequency of $Z_H$ generated





with standard deviation of 7.0 dBZ and mean of 30.0 dBZ. Fig. 12(b) shows the
corresponding occurrence frequency of $Z_{DR}$ with 0.5 dB standard deviation and 1.0 dB mean.
Fig. 12(c) shows the occurrence frequency of $K_{DP}$ generated with $0.5°$ $km^{-1}$ standard
deviation and $1.0°$ $km^{-1}$ mean. Fig. 12 (d) shows the occurrence frequency of $A_H$ generated
with $0.01°$ $km^{-1}$ standard deviation and $0.0003°$ $km^{-1}$ mean.
To investigate the extent of contamination of the rainfall amount by propagation of errors in
each polarimetric variable for $R(Z,Z_{DR},K_{DP},A_H)$, the errors of Z, $Z_{DR}$, and $K_{DP}$ ingested to
simulated data were 0 to 5 dBZ with interval 0.25 dBZ, 0 to 0.6 dB with interval 0.03 dB, and
0 to 0.2 degree $km^{-1}$ with interval 0.01 degree $km^{-1}$, respectively. The rain rate was calculated
by same $R(Z,Z_{DR},K_{DP},A_H)$ as applied to real data in the previous Sect.. The RMSE and NE
were calculated for rainfall amount with and without error-ingested polarimetric variables.
The rainfall amount obtained using the raw simulated variables was used as a reference.
Fig. 13 shows the RMSE and NE distribution of different polarimetric rainfall relations with
ingested error. The magenta, black, red, green, blue, and purple lines show RMSE and NE
obtained by the rainfall relations $R(Z)$, $R(K_{DP})$, $R(Z,K_{DP},A_H)$, $R(Z,Z_{DR})$, $R(K_{DP},Z_{DR})$, and
$R(Z,Z_{DR},K_{DP},A_H)$, respectively. The threshold rainfall was from 0 to 300 mm $h^{-1}$ for
calculating statistical scores. Fig. 13(a) shows the RMSE distribution of each rainfall relation
with different ingested error step. The RMSE of $R(Z,K_{DP},A_H)$ is the largest of all the rainfall
relations. The RMSE of $R(Z,Z_{DR},K_{DP},A_H)$ is higher than that of $R(Z)$, $R(Z,Z_{DR})$, and
$R(K_{DP},Z_{DR})$ but less than that of $R(K_{DP})$. It means that not all errors from Z, $Z_{DR}$, and $K_{DP}$
propagate into the $R(Z,Z_{DR},K_{DP},A_H)$. Fig. 13(b) shows the corresponding distributions for NE.
The value of NE increases in the order $R(Z,Z_{DR},K_{DP},A_H)$, $R(Z,K_{DP},A_H)$, $R(K_{DP})$, $R(K_{DP},Z_{DR})$,
$R(Z,Z_{DR})$, and $R(Z)$. In Sect. 3.3 and 4.1, the statistical scores of $R(Z,Z_{DR},K_{DP},A_H)$ did not
change significantly with respect to different $Z_H$ and $Z_{DR}$ biases. The results of the simulation
and observations suggest that the accuracy of $R(Z,Z_{DR},K_{DP},A_H)$ is relatively weakly affected
by errors in each polarimetric variable.
**5   Conclusions**
To improve polarimetric rainfall estimation and examine the candidates for an optimum
rainfall relation using polarimetric variables observed from the Bislsan radar, the first
polarimetric radar in Korea, a heavy rainfall case of 7 hours duration caused by low-pressure
conditions on 25 August 2014 was analysed.





The theoretical approach to investigate the observed $Z_{DR}$ quality used the relation between the
standard deviation of $Z_{DR}$ and $\rho_{hv}$ using the scan strategy parameters of the Bislsan radar.
The result showed that more samples were required to achieve the theoretical accuracy in $Z_{DR}$.
The best performance was obtained when a moving average $Z_{DR}$ with window size of 9 gates
was applied to the rainfall estimation using $R(Z,Z_{DR})$ and to the calculation of $Z_H$ bias. The
$Z_{DR}$ quality check should be performed before using $Z_{DR}$ for quantitative applications like
rainfall estimation and hydrometeor classification for the Bislsan radar. We also expect that
the light rain method for obtaining the $Z_{DR}$ bias may be used as an alternative to the vertical
pointing scan method, because the rainfall estimation using this method performed well in our
case. Using DSD data for the calculation of $Z_{DR}$ bias might give more accurate rainfall
estimation with $R(Z,Z_{DR})$.
Finally, the accuracy of $R(Z,Z_{DR},K_{DP},A_H)$ was not very sensitive to $Z_{DR}$ and $Z_H$ biases in both
observations and simulations. Thus $R(Z,Z_{DR},K_{DP},A_H)$ is expected to be less sensitive to $Z_{DR}$
and $Z_H$ errors and could be used to estimate rainfall for heavy rainfall cases in Korea until an
accurate hydrometeor classification algorithm is developed.
**Acknowledgements**
The authors acknowledge the Ministry of Land, Infrastructure, Transport and the Korea
Meteorological Administration for providing radar data and AWS data for this work. The
authors acknowledge Prof. V. N. Bringi at Colorado State University, who provided the
scattering simulation code. The authors also acknowledge Prof. A. C. Illingworth at Reading
University who provided valuable comments on $Z_{DR}$ data quality. This research was funded
by the Korea Meteorological Industry Promotion Agency under Grant KMIPA 2015-1050.





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





1   Table 1. Polarimetric radar rainfall relations used in this study.

| Relations | | Relations | |
|---|---|---|---|
| $R(Z)$ | $R=0.017Z^{0.714}$ | $R(K_{DP})$ | $R=61.5K_{DP}^{0.908}$ |
| $R(A_H)$ | $R=3409A_H^{1.02}$ | $R(Z,Z_{DR})$ | $R=0.0148Z^{0.818}Z_{DR}^{-3.72}$ |
| $R(K_{DP},Z_{DR})$ | $R=82.2K_{DP}^{0.855}Z_{DR}^{-1.977}$ | $R(Z,K_{DP},A_H)$ | $R=17211Z^{-0.027}K_{DP}^{0.62}A_H^{0.65}$ |
| $R(Z,Z_{DR},K_{DP},A_H)$ | | $R=4502Z^{-0.014}Z_{DR}^{-0.389}K_{DP}^{0.486}A_H^{0.653}$ | |

3   Table 2. Statistics of the comparison of hourly rainfall amount between $R(Z,Z_{DR})$ and gages.

| Relation | $Z_H$ bias source | $Z_{DR}$ bias source | RMSE | NE | CC |
|---|---|---|---|---|---|
| $R=0.0148Z^{0.818}Z_{DR}^{-3.72}$ | Observed $Z_{DR}$ | Observed $Z_{DR}$ | 17.2 | 0.66 | 0.77 |
| | | $mZ_{DR}$ | 9.2 | 0.56 | 0.95 |
| | $mZ_{DR}$ | Observed $Z_{DR}$ | 15.2 | 0.53 | 0.77 |
| | | $mZ_{DR}$ | 7.4 | 0.45 | 0.95 |

5   Table 3. Same as Table 2 but for $R(Z,Z_{DR},K_{DP},A_H)$.

| Relation | $Z_H$ bias source | $Z_{DR}$ bias source | RMSE | NE | CC |
|---|---|---|---|---|---|
| $R=4502Z^{-0.014}Z_{DR}^{-0.389}K_{DP}^{0.486}A_H^{0.653}$ | Observed $Z_{DR}$ | Observed $Z_{DR}$ | 5.2 | 0.30 | 0.95 |
| | | $mZ_{DR}$ | 5.2 | 0.30 | 0.95 |
| | $mZ_{DR}$ | Observed $Z_{DR}$ | 5.3 | 0.30 | 0.95 |
| | | $mZ_{DR}$ | 5.4 | 0.31 | 0.95 |





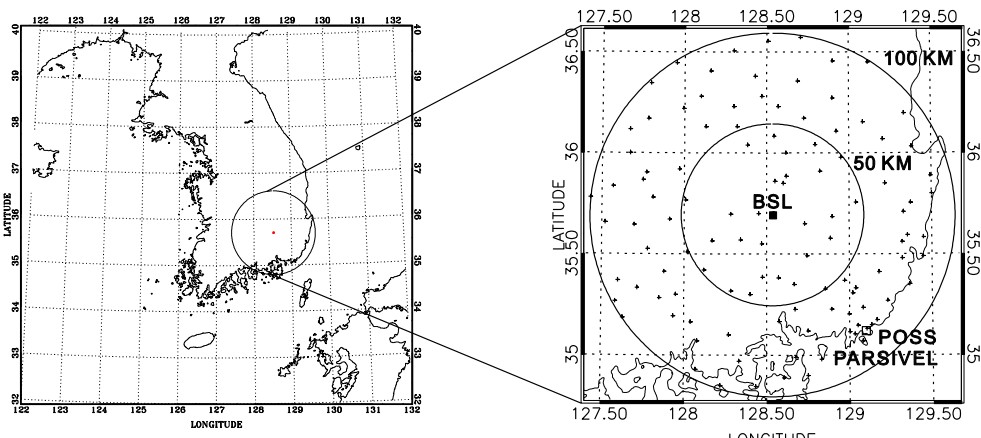

Figure 1. Location of Bislsan radar (solid rectangle), the POSS and PARSIVEL disdrometer
(open rectangle), and rain gages (plus signs) distributed within 100 km of the radar. The
circles are at 50 and 100 km from the radar.





1                        (a)

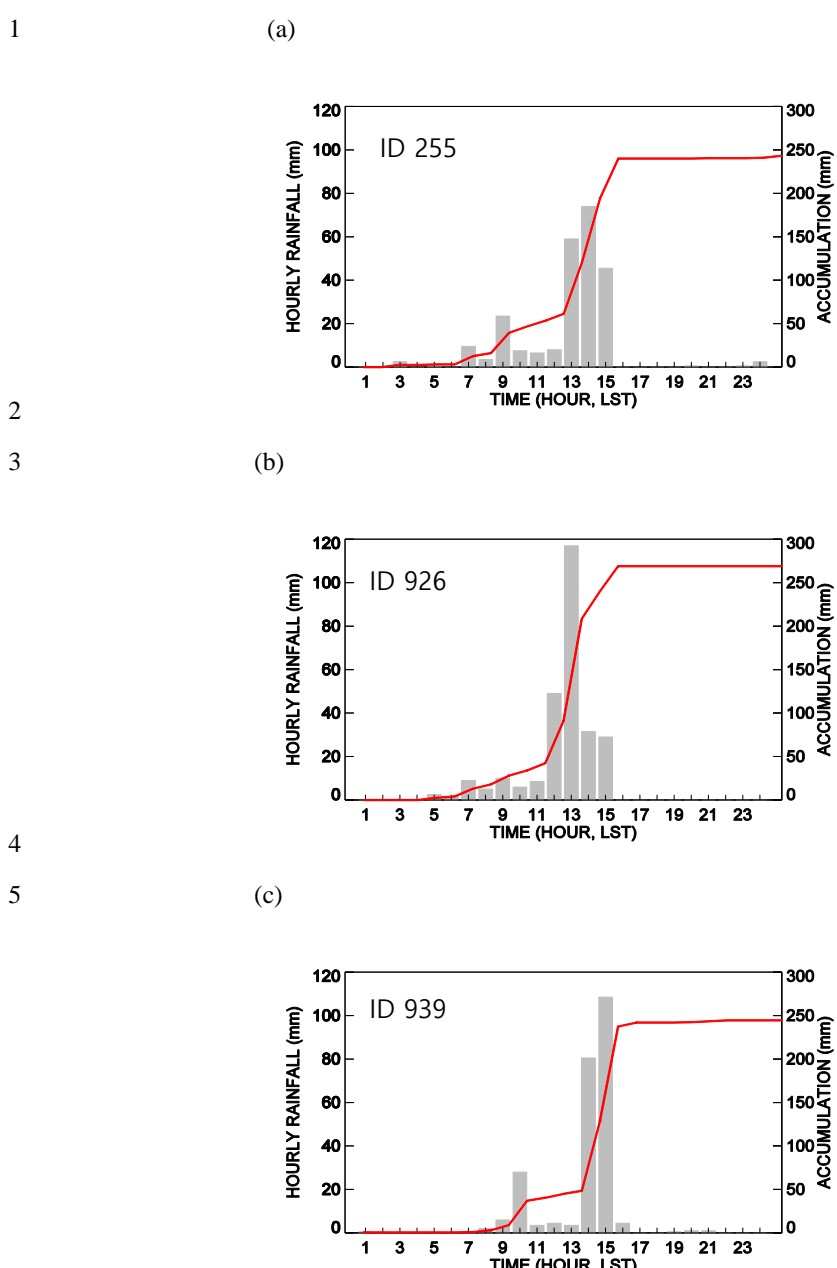

3                        (b)

5                        (c)

7        Figure 2. Time series of hourly rainfall (gray bars) and accumulation (red line) from the three

8        gages that recorded the highest rainfall (a) ID 255, (b) ID 926, (c) ID 939.





1                  (a)

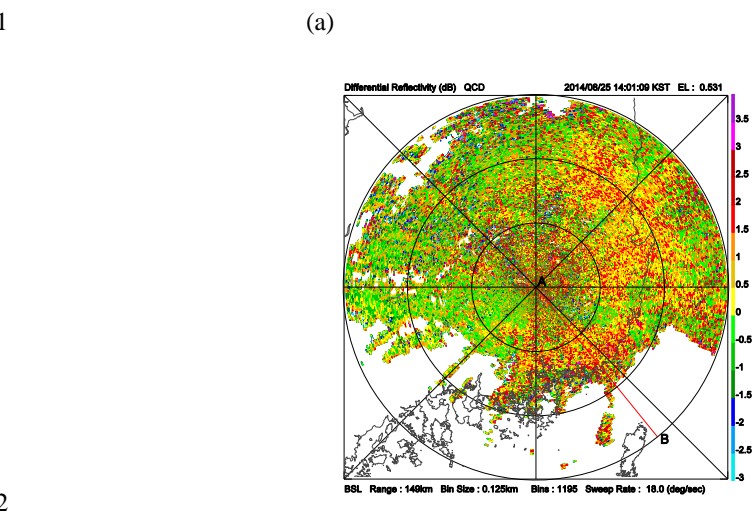

4          (b)                                        (c)

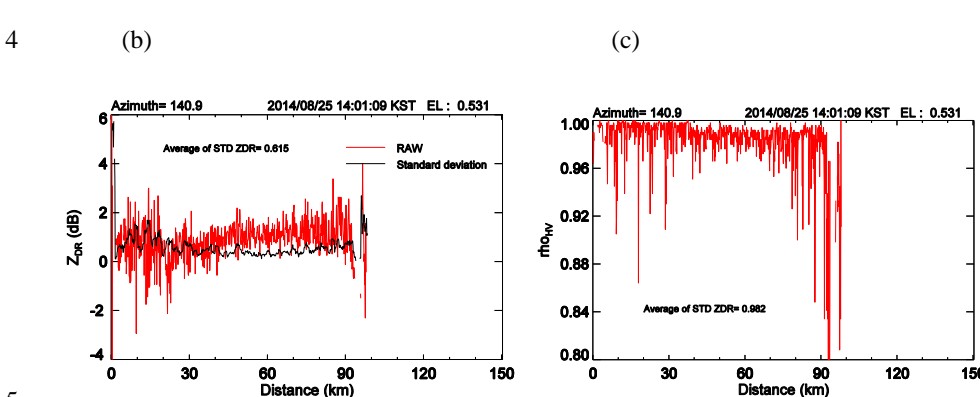

Figure 3. (a) Distribution of $Z_{DR}$ within the radar coverage, (b) the radial profile of $Z_{DR}$ and (c)

7       cross correlation along the line A–B in (a) at 1401 LT 25 August 2014.





(a)                (b)

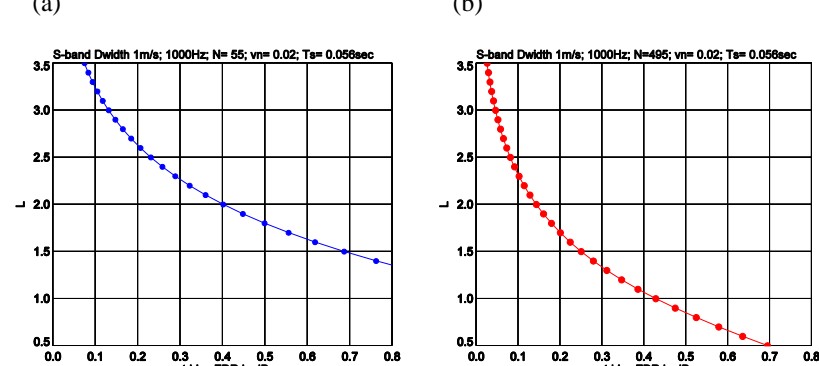

Figure 4. Theoretical relation between standard deviation of $Z_{DR}$ and cross correlation using
(a) the scan configuration of the Bislsan radar. Dwell time is 56 ms, number of samples is 55,
normalized Doppler width is 0.02; (b) same as (a) but for 495 samples.





1                          (a)

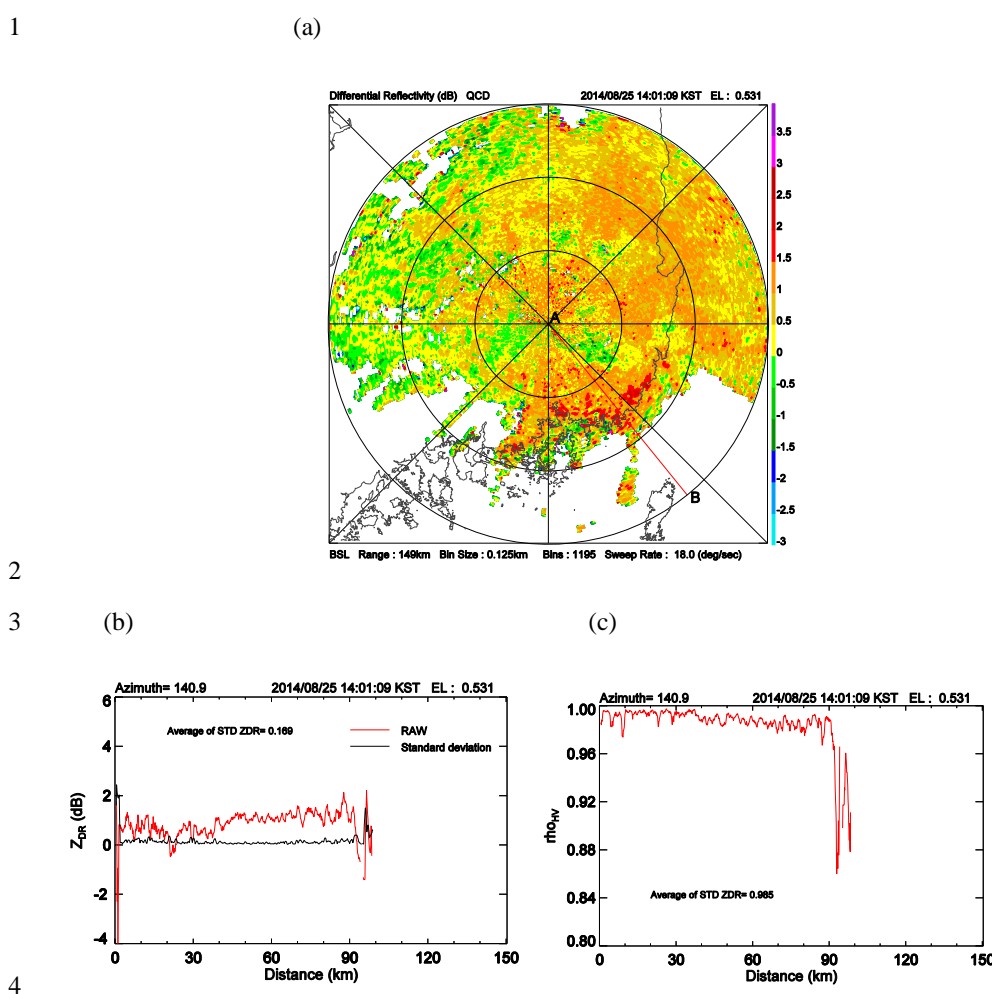

3      (b)                                         (c)

5     Figure 5. Same as Fig. 3 but for moving averages of $Z_{DR}$.



(a)                                (b)

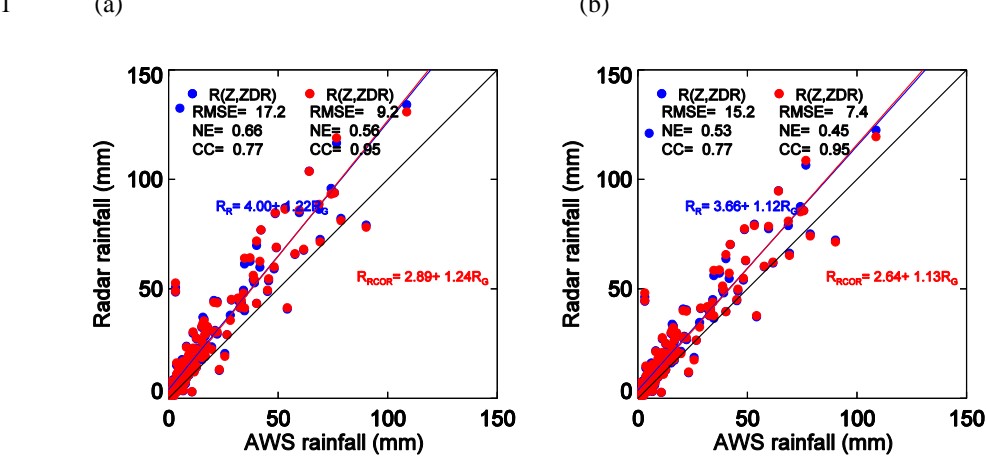

Figure 6. Scatter plot of 1 hour rainfall obtained by R(Z,Z$_{DR}$) against gage rainfall. (a) Radar
rainfall was calculated using Z$_H$ bias calculated from observed Z$_{DR}$ bias and Z$_{DR}$ biases
calculated from observed Z$_{DR}$ (blue full circles) and mZ$_{DR}$ (red full circles), (b) same as (a)
but for Z$_H$ bias calculated from mZ$_{DR}$.
(a)                                (b)

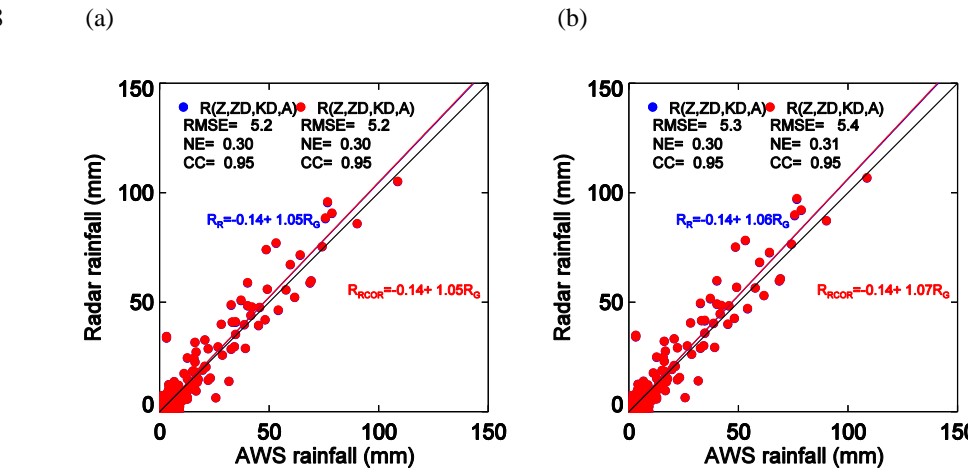

Figure 7. Same as Fig. 6 but for radar rainfall obtained by R(Z,Z$_{DR}$,K$_{DP}$,A$_H$).





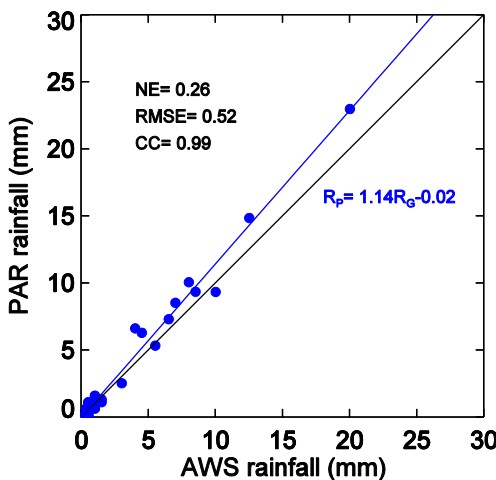

Figure 8. Scatter plot of 10 min rainfall amount measured by PARSIVEL and gage for 24
hours.

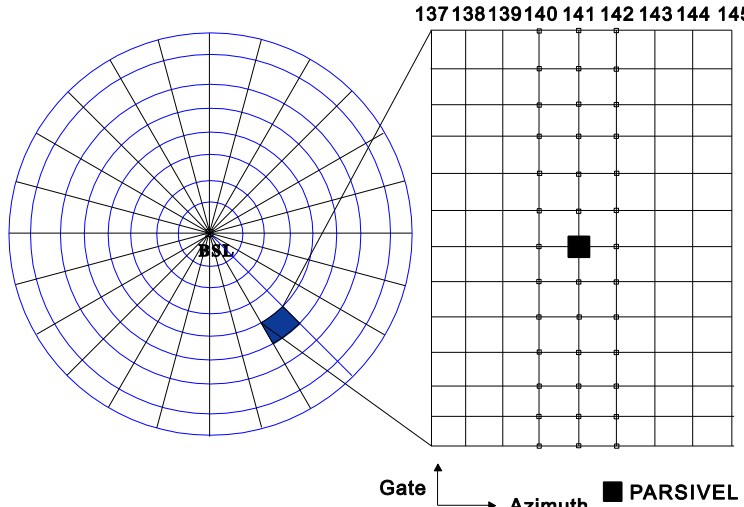

Figure 9. Schematic diagram for the comparison of radar and PARSIVEL $Z_{DR}$. The numbers
refer to azimuth angle.





1          (a)                                              (b)

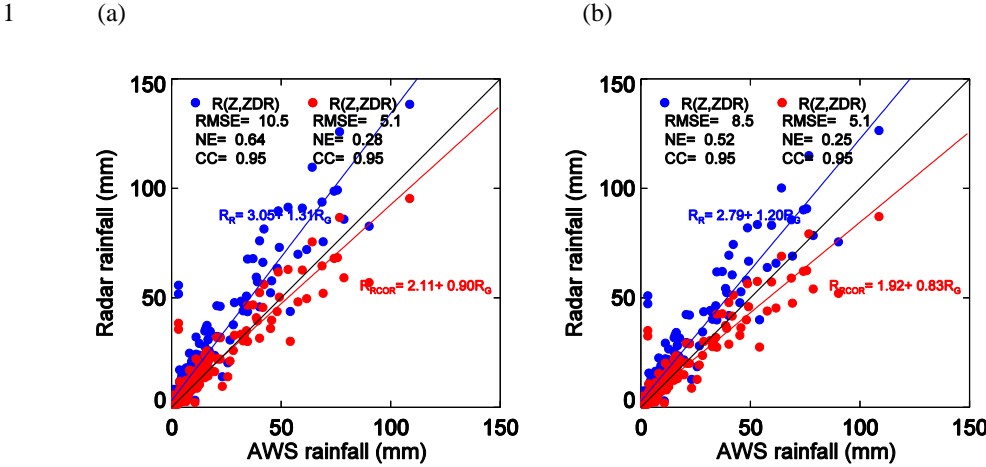

3     Figure 10. Same as Fig. 6 but for $Z_{DR}$ biases determined by comparison between radar and

4        PARSIVEL.

6          (a)                                              (b)

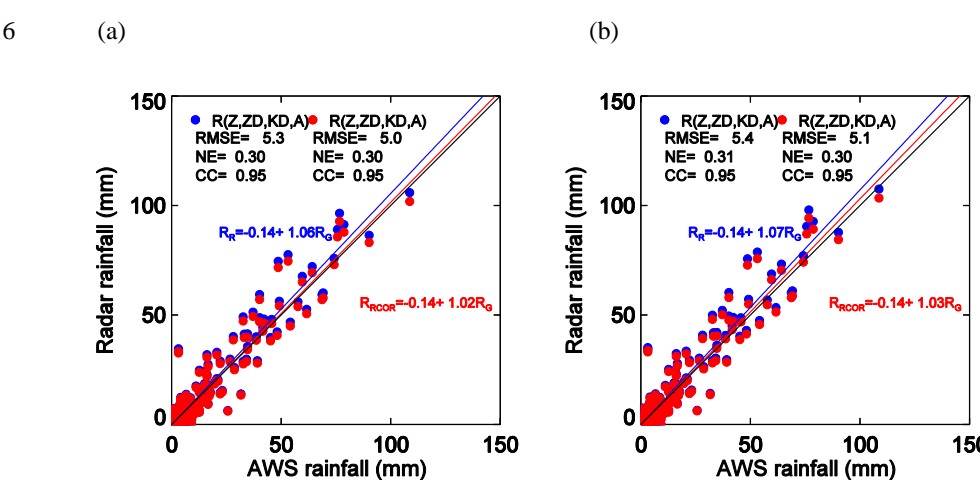

8     Figure 11. Same as Fig. 7 but for $Z_{DR}$ biases determined by comparison between radar and

9        PARSIVEL.





1          (a)                               (b)

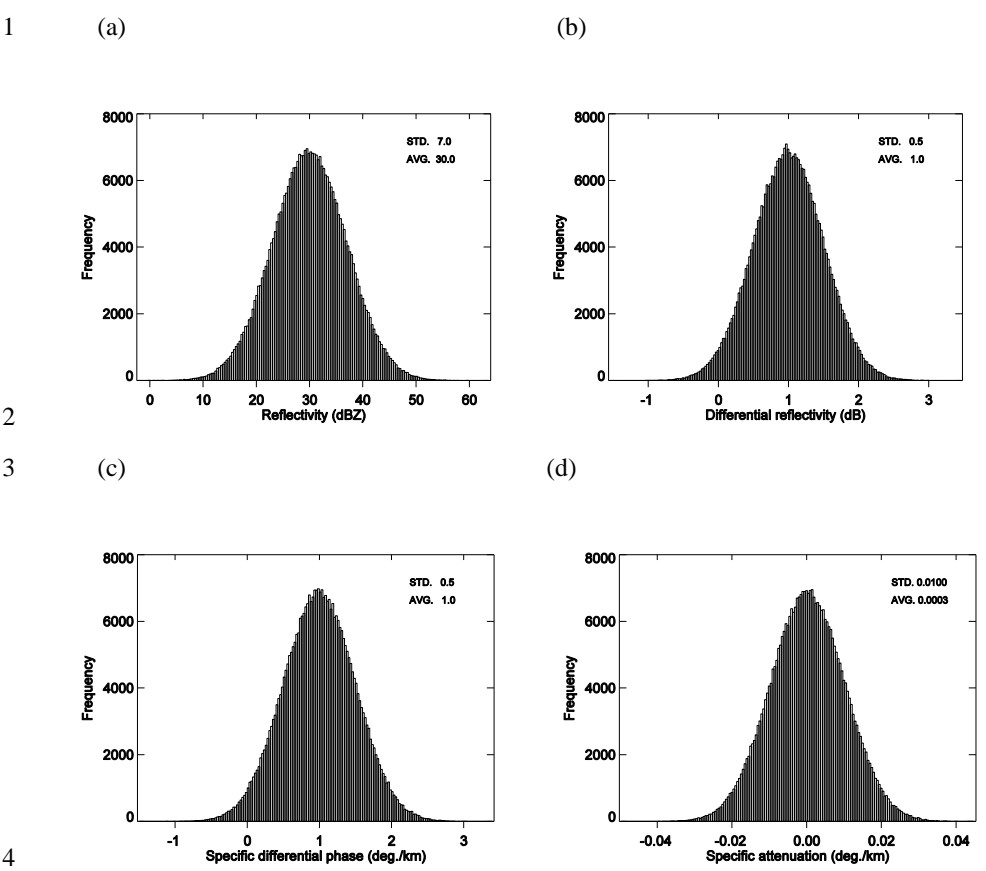

3          (c)                               (d)

5          Figure 12. Distribution frequencies of the generated polarimetric variables. (a) $Z_H$, (b) $Z_{DR}$, (c)

6              $K_{DP}$, and (d) $A_H$.





(a)

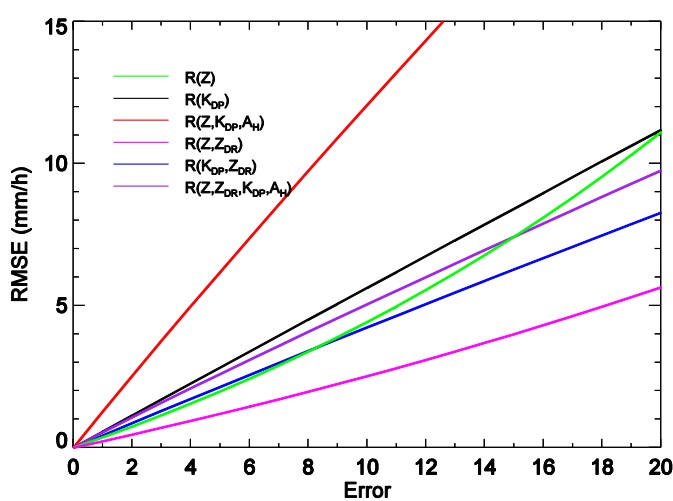

(b)

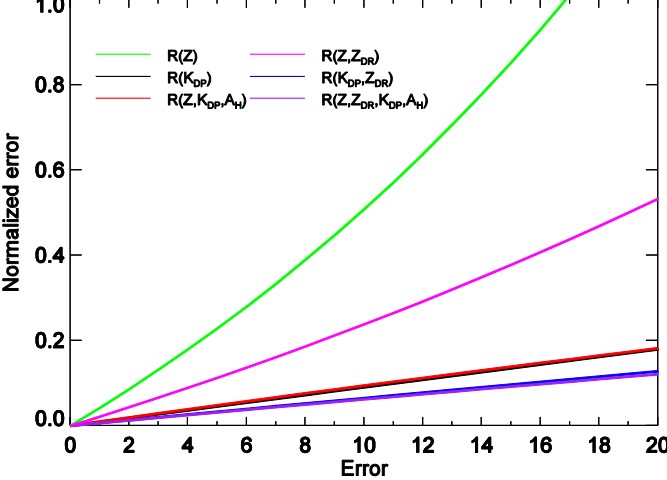

Figure 13. Distribution of (a) RMSE and (b) NE with generated errors for different rainfall
relations. Magenta, black, red, green, blue, and purple lines show RMSE and NE obtained
from the rainfall relations R(Z), R($K_{DP}$), R(Z,$K_{DP}$,$A_H$), R(Z,$Z_{DR}$), R($K_{DP}$,$Z_{DR}$), and
R(Z,$Z_{DR}$,$K_{DP}$,$A_H$), respectively.