# Peer review of "The possibility of rainfall estimation using $R(Z,Z_{DR},K_{DP},A_{H})$"

_Hydrology and Earth System Sciences, 2015_

## Referee Comment (RC1) · Anonymous Referee #1 · 15 Feb 2016

Title: The possibility of rainfall estimation using R(Z,ZDR,KDP,AH) : A case study of heavy rainfall on 25 August 2014 in Korea Author(s): C.-H. You et al. MS No.: hess-2015-515 MS Type: Research article

General Comments The authors attempt to validate a dualpol rain rate estimator. This should be stated in the title. To use "possibility" is the wrong term. The authors touch important aspects and have some interesting ideas: finding a correct rr estimator, addressing the sampling issues for dualpol data, but in my view none of the issues are analyzed to an extent that is needed. For example ZDR. A method to derive the ZDR bias from disdrometer data is suggested. But I cannot find a verification of this approach. Also, the paper needs to be restructured. As such, the ZDR aspects are

spread throughout the paper. So everything related to data quality should be one section (especially with respect to ZDR). Once this is handled, one can focus on the rain rate estimator, where the proposed one is compared to the ones that are typically used. In the current form it is really hard to collect all the pieces of information. The authors just state that there is no birdbath Data available for this event. The way they write the paper, it appears that the system is capable doing so. Then it should be an easy task to assess the proposal with a different set of data. I think there is material for a paper but the current form is not sufficient for publication.

Specific comments: The authors suggest that ZDR from a disdrometer might be used for calibration. This implies that ZDR is constant with height, so that such an estimate can be related to the radar height. This might be true to some extent in an stratified rain situation, but the rain rates considered here are certainly beyond a stratiformed rain situation. In other words the argument here is missing a proof, at least a discussion of possible limitations is needed. The authors do not attempt to make an recommendation which rr to use now. Or is there further research needed? What about the "robustness" of the approach, s.th. the authors state in the introduction? This is not addressed.

What is the accuracy goals for the estimator. How did the rr estimator work for other data sets? Anyone else is using it? What accuracy is achieved there? Ah (I assume the path integrated attenuation) is not introduced. How is it computed, quality control, formula. Also the rr estimators should be introduced in more detail. For some of the estimators recommendations can be found in literature. Why not using a R(kdp) relationship.? The rr estimators need to be introduced in the text (not just in a table) with references.

Did the authors verify the result with an other data set? Section 2.1.

Quality control of disdrometer: very crude. It is well known that wind effects can bias disdrometer measurements. Did the authors check for this? If not, they should do this and figure out how the conclusions may change in their study.

[Figure]

Section 3.1

SD (ZDR): here terms accuracy and bias seem not properly separated. What a bias/ uncertainty is should clearly separated.

Error propagation (4.2):

not really introduced here. How are the distributions used? Independently (like a Monte-Carlo simulation). Here, suddenly other estimators are discussed here (R(Kdp) for example). For the real world comparision, these are not discussed. Why? If you want focus on R(Z/ZDR) and the proposed one here, you focus on these here as well. Or you say s.th abouth the performance (statistically) in section 3.3.

p 12/ l. 12 ff: So if you have a HMC, you will abandon your proposed rain estimator? Really? So you implicitely state here, that it is not the best choice. What is a better choice??

Technical correction a more detailed review is need after the paper is fully revised.

p157/l. 28: Diederich

---

## Author Comment (AC1) · 19 Feb 2016

Response to review : At first, thank you very much for referee's effort in reviewing our paper even your busy time. We revised the manuscript titled "The possibility of rainfall estimation using R(Z,ZDR,KDP,AH): A case study of heavy rainfall on 25 August 2014 in Korea" that was submitted to Hydrology and Earth System Sciences. The manuscript has been revised as suggested. We would appreciate any feedback on the revisions. Response to review by Anonymous referee #1

General comments 1. The authors attempt to validate a dualpol rain rate estimator. This should be stated in the title. To use "possibility" is the wrong term.

[Figure]

Author's Response :

We changed the title to "The validation of polarimetric rainfall estimators using R(Z,ZDR) and R(Z,ZDR,KDP,AH) in Korea" as reviewer's comment.

2. The authors touch important aspects and have some interesting ideas: finding a correct rr estimator, addressing the sampling issues for dualpol data, but in my view none of the issues are analyzed to an extent that is needed. For example ZDR. A method to derive the ZDR bias from disdrometer data is suggested. But I cannot find a verification of this approach.

Author's Response :

Thank you for your comment. The reference of ZDR bias measured by vertical pointing scan was needed to verification of ZDR bias obtained by this approach. The scan strategy of the BSL radar operated by Ministry of Land, Infrastructure, Transportation is not included vertical pointing scan for their purpose (rapid update data). That is why we are trying to find out the method for calculation of good ZDR bias and qualified ZDR. Therefore, we tried to validate the approach using rainfall comparison of radar and gage. Of course, there are many uncertainty for the estimation of radar rainfall such as difference of gage height and radar beam height, accuracy of radar rainfall relations, etc..

3. Also, the paper needs to be restructured. As such, the ZDR aspects are spread throughout the paper. So everything related to data quality should be one section (especially with respect to ZDR). Once this is handled, one can focus on the rain rate estimator, where the proposed one is compared to the ones that are typically used. In the current form it is really hard to collect all the pieces of information.

Author's Response:

We re-organized the manuscript as reviewer's comment. In "3. Result", we divided it two parts "3.1 Data quality of ZDR" composed of "3.1.1 Improvement of ZDR data

quality using moving average", "3.1.2 Absolute bias correction of ZDR and ZH", and "3.1.3 Improvement of ZDR data quality using disdrometer" and "3.2 Validations of rainfall relations" including "3.2.1 The performance of rainfall relations with different ZH and ZDR biases obtained from the observed ZDR and mZDR", "3.2.2 The performance of the relations using ZDR bias obtained by disdrometer", "3.2.3 The simulation of R(Z,ZDR,KDP,AH) using generated variables". We modified figure number and related descriptions accordingly.

4. The authors just state that there is no birdbath data available for this event. The way they write the paper, it appears that the system is capable doing so. Then it should be an easy task to assess the proposal with a different set of data. I think there is material for a paper but the current form is not sufficient for publication.

Author's Response:

Thanks to reviewer's comment, we could add two more cases in the manuscript. The three rainfall events occurred on 23 August 2012, 8 September 2012, and 25 August 2014, which were caused by indirect effect of Typhoon, low pressure accompanied with the front, and low pressure were included and summarized in Table 1. We added related figures and description in the manuscript and also modified the some mistakes

Specific comments

1. The authors suggest that ZDR from a disdrometer might be used for calibration. This implies that ZDR is constant with height, so that such an estimate can be related to the radar height. This might be true to some extent in an stratified rain situation, but the rain rates considered here are certainly beyond a stratiformed rain situation. In other words the argument here is missing a proof, at least a discussion of possible limitations is needed.

Author's Response:

We agree that there is some limitations of using disdrometer data especially for the

convective systems which have much fluctuation of DSD with height. There would be fluctuation of DSD with height in three cases we analyzed. Unfortunately, there was no way to find out the distribution. As reviewer's comment, we added the limitations in the Sect. 4. Conclusions of the manuscript as follows; "Using DSD data for the calculation of ZDR bias might give more accurate rainfall estimation with R(Z,ZDR), even it is limited to the homogeneous DSD at the layer between radar beam height and ground and not strong wind condition which could degrade the quality of ZDR calculation from disdrometer."

2. The authors do not attempt to make an recommendation which rr to use now. Or is there further research needed? What about the "robustness" of the approach, s.th. the authors state in the introduction? This is not addressed.

Author's Response:

Thank you very much for your comments. The study is of course needed to obtain more and more accurate radar rainfall estimation. In this study, we found out that R(Z,ZDR) was better when the ZH bias and ZDR bias can be obtained accurately otherwise R(Z,ZDR,KDP,AH) had better comparing with both relations. We added the description in Sect. 4 of the manuscript as follows; "Using DSD data for the calculation of ZDR bias might give more accurate rainfall estimation with R(Z,ZDR), even it is limited to the homogeneous DSD at the layer between radar beam height and ground and not strong wind condition which could degrade the quality of ZDR calculation from disdrometer. Comparing the statistical scores between the most accurate R(Z,ZDR) and R(Z,ZDR,KDP,AH) in this study, R(Z,ZDR) had better performance than that of R(Z,ZDR,KDP,AH). However, R(Z,ZDR,KDP,AH) is expected to be less sensitive especially to ZH and ZDR errors in both observations and simulations. Therefore, R(Z,ZDR,KDP,AH) could be used as a representative rainfall relation in case ZDR bias was not calculated accurately in Korea."

3. What is the accuracy goals for the estimator. How did the rr estimator work for other

data sets? Anyone else is using it? What accuracy is achieved there?

Author's Response:

We think it is related to number 2 of reviewer's comment. We found out that R(Z,ZDR) was better when the ZH bias and ZDR bias can be obtained otherwise R(Z,ZDR,KDP,AH) would better in this study.

4. Ah(I assume the path integrated attenuation) is not introduced. How is it computed, quality control formula.

Author's Response:

Reviewer's is totally right. We used following method which was explained in You et al. (2015a) to calculate AH from polarimetric radar. We added the some descriptions for AH calculation in the Sect. 2.2 of manuscript like "AH was calculated from the radial profile of the attenuated reflectivity and two-way PIA (Path Integrated Attenuation) along the propagation path using observed ZH, differential phase shift from BSL radar. The more detailed description for AH calculation can be found in You et al. (2015a)." Please refer to supplement 1.

5. Also the rr estimators should be introduced in more detail. For some of the estimators recommendations can be found in literature. Why not using a R(Kdp) relationship?

Author's Response:

As reviewer may understand our purpose of this study, we would like to focus on the performance of polarimetric radar rainfall relations with respect to ZDR. That is why we compared two relations R(Z,ZDR) and R(Z,ZDR,KDP,AH) which are dependent on ZDR data. For reviewer's understanding, the results from R(KDP) were shown below.

Figure 1. Scatter plots of gage rainfall and R(KDP) on (left) Case 1, (middle) Case 2, (right) Case 3.

6. The rr estimators need to be introduced in the text (not just in a table) with references.

Author's Response:

We added following sentences in Sect. 2.3 of the manuscript. Fig. 3 shows the scatter plot of rain rate obtained from disdrometer and polarimetric radar rainfall relations R(Z,ZDR) and R(Z,ZDR,KDP,AH). The CC and RMSE of R(Z,ZDR) (R(Z,ZDR,KDP,AH)) were 0.95 (0.99) and 3.5 mm h-1 (1.2 mm h-1) which are same relations used in You et al. (2015a). The rainfall relations for validation and simulation are summarized in Table 2.

7. Did the authors verify the result with another data set? Section 2.1.

Author's Response:

As reviewer's comments, we added two more cases in the manuscript as mentioned in above answer to the reviewer's comments related to data set.

8. Quality control of disdrometer: very crude. It is well known that wind effect can bias disdrometer measurements. Did authors check for this? If not, they should do this and figure out how the conclusions may change in their study.

Author's Response:

Thank you for your comments. It is true that wind effect can give bias disdrometer measurements. Actually, we did not consider wind effect for this study. We did quality control algorithm using the equation suggested by Jaffrain and Berne (2011) to reduce the wind effect. |ãĂŰv(D)ãĂŮ_meas-ãĂŰv(D)ãĂŮ_Beard |≤0.6ãĂŰv(D)ãĂŮ_Beard Where, v(D)meas is the velocity measured by PARSIVEL, v(D)Beard is the velocity for a drop diameter D according to Beard's model. However, to know the accuracy of disdrometer, 10 mins rainfall amount measured from disdrometer and gage were compared. We found out that rain rate obtained from disdrometer was correlated to that of gage as shown in Fig. 7 in the manuscript. We checked the maximum wind speed for all cases were less than 8 ms-1. Friedrich and Higgings (2013) found out

[Figure]

that once the wind speed exceeded a critical value, approximately 15-20 ms-1 based on the observations during Hurricane Ike and VORTEX2, the PARSIVEL continuously observed unrealistically large slow-falling drops as seen during Hurricane Ike. Therefore, we thought that the disdrometer data can be used for the analysis. Anyway, we added the limitations in the Sect. 3.2.2 of the manuscript as follows; "However, these results would be changed when the drop size distribution of the rainfall system was fluctuated with height, especially at the layer between radar beam and ground. And the wind effect is another limitation of this results."

Figure 2. Time series of the 10min. maximum wind speed in one hour from Automatic Weather System located nearby PARSIVEL for three cases

References: Jaffrain, J., Berne, A.: Experimental quantification of the sampling uncertainty associated with measurements from PARSIVEL disdrometers, J. Hydrometeor., 12, 352-270, 2011. Friedrich, K., Higgings S.: Articulating and stationary PARSIVEL disdrometer measurements in conditions with strong winds and heavy rainfall, J. Atmos. Ocean. Technol., 30, 2063-2080.

Section 3.1 SD(ZDR): here terms accuracy and bias seem not properly separated. What a bias/uncertainty is should clearly separated.

Author's Response:

We changed "accuracy" into "performance" in the manuscript.

Error propagation (4.2): Not really introduced here. How are the distributions used? Independently (like a Monte-Carlo simulation). Here, suddenly other estimators are discussed here(R(Kdp) for example). For the real world comparison, these are not discussed. Why? If you want focus on R(Z,ZDR) and the proposed one here, you focus on these here as well. Or you say s.th about the performance (statistically) in section 3.3.

Author's Response:

In this section, we would like to show effect of the multi-cor-linearity for R(Z,ZDR,KDP,AH) because ZH and KDP are highly correlated as shown below Table 1. This can reduce the accuracy of relation for some cases but not all cases. To understand the robustness of the relation, we generated polarimetric variables just assuming Gaussian distribution as mentioned in the manuscript. The simulation and observation showed the ZH and KDP could interact in the direction of reducing an error of the relation from the simulation.

Table 1. The correlation coefficients of variables measured by POSS disdrometer. (Kang, 2016) : please Fig. 3 of attaches.

P 12/l. 12 ff: So if you have a HMC, you will abandon your proposed rain estimator? Really? So you implicitly state here, that it is not the best choice. What is a better choice?

Author's Response:

Thank you for your comments. We just would like to mention that both relations used in this study is not adequate for the solid precipitation like snow, hail. Anyway, we changed the sentences in the Sect. 4 as follows; "R(Z,ZDR,KDP,AH) is expected to be less sensitive especially to ZH and ZDR errors in both observations and simulations. Therefore, R(Z,ZDR,KDP,AH) could be used as a representative rainfall relation in case ZDR bias was not calculated accurately in Korea."

\*\*\* Thank you very much again for your deep review and it will be of much help for better our manuscript quality.\*\*\*

Please also note the supplement to this comment:
http://www.hydrol-earth-syst-sci-discuss.net/hess-2015-515/hess-2015-515-AC1-supplement.zip

Figure 1. Scatter plots of gage rainfall and R($K_{DP}$) on (left) Case 1, (middle) Case 2, (right) Case 3.

**Fig. 1.**

Figure 2. Time series of the 10min. maximum wind speed in one hour from Automatic Weather System located nearby PARSIVEL for three cases.

**Fig. 2.**

Table 1. The correlation coefficients of variables measured by POSS disdrometer. (Kang, 2016)

| Variables | ZH | ZDR | KDP | AH |
|-----------|------|------|------|------|
| ZH | 1 | 0.31 | 0.98 | 0.69 |
| ZDR | 0.31 | 1 | 0.28 | 0.16 |
| KDP | 0.98 | 0.28 | 1 | 0.69 |
| AH | 0.69 | 0.16 | 0.69 | 1 |

**Fig. 3.**

---

## Referee Comment (RC2) · Anonymous Referee #2 · 23 Feb 2016

Overview

The aim of the authors is to find the optimum quantitative precipitation estimation method for the first dual-pol S-Band radar in Korea. To this end they search for the best method to remove biases in (horizontally polarized) reflectivity measurements ($Z_h$) and in differential reflectivity ($Z_{DR}$). They test mainly $R(Z_h, Z_{DR})$ and $R(Z_h, Z_{DR}, K_{DP}, A_h)$, the later one proves to provide quite stable and reliable estimates of the rain intensity.

A basic requirement for a publication is, that an (educated) reader is able to comprehend what was done and that she/he is able to repeat the investigation, based on the

information of the publication. This goal is not met by this manuscript.

I did not find a clear structure within the paper. It should be stated clearly, that different approaches to determine the biases in $Z_h$ and $Z_{DR}$ will be discussed in advance of describing the first approach. The abstract does not describe what is done in this investigation.

The paper contains a large amount of oversight, reducing the confidence in the care the authors took.

The scientific innovation within this paper is quite limited. Known QPE approaches are tested and the results gathered during a strong precipitation event are presented. It is questionable if this is innovative enough to justify publication. Because of the limited quality in presentation I would reject the paper in the present form.

Specific Notes

- p2, l16: "different drop shape", different from what?

- p2, l28: KMA installed an S-Band polarimetric radar in the fare northwest of Korea. Later in the text, the Bislsan radar was the first polarimetric radar in Korea. Bislsan is in the southeast of Korea. Is this a contradiction or are there at least two polarimetric radars in Korea?

- p3, l26: Fig. 1 show the location of all instruments? Where are the rain gages mentioned at the beginning of chapter 2.3?

- p4, l2: Radar Bislsan is (according to my digital elevation model) at a height of more than 1 km asl. The distrometers are quite close to see level. In 82 km from radar the $0.5°$ beam is 1.1 km above radar height. There is nearly 2 km separation between radar and distrometer measurements? Are these data comparable? You should at least discuss this problem.

- p4, l12f: Drop numbers count only in the lower channels leads to an removal of the data. In the next step you remove drop size spectra with drop number counted only in the lower 5 channels? This is done twice.

- p4, l25: The prefactor of D is 0.00057, not 0.5. . . .

- p5, l24f: What is N? N is the number of rain gages (121) or the number of hourly measurements ($7 \times 121$)?

- p6, l20: There is no Bringi and Chandrasekar (2003). You assumably meant 2001, or did you mean Bringi et al. 2003?

- p6, l21: At least in Bringi and Chandrasekar (2001) I did not find this equation. Please give a more precise citation.

  What is $\rho_{co}$, what is $\rho_{co}(l)$? Is $\rho_{co} = \rho_{co}(0)$?

  You call the the correlation at different places $\rho_{co}$, $\rho_{co}(l)$, $\rho[n]$ and $\rho_{hv}$. Is it all the same thing? So please use the same notation. Are these different terms? So please indicate what is meant by which term.

- p7, l6: With equation 8, an L of over 3 (line 11) is reached by $\rho_{hv} > 0.5$ and an L of 1.7 (line 14) needs $\rho_{hv} = .32$. Probably the prefactor 10 is wrong in equation 8.

- p8, l2: Ryzhkov et al. (2005a)

- p8, l19: Why do you have problems to detect the melting layer by a 6 elevation volume scan? There are approaches to determin the melting layer from an individual elevation.

- p10, l24: Table 3 contains the results from chapter 3.3. The results from chapter 4.1 are not given.

- p11, l5: As far as I got it, you never introduced $A_h$, although that term is already used in the title. I asume, $A_h$ is the path integrated attenuation for the horizontally polarized wave. This should be measured in dB/km but not in degrees/km.

- p11, l18: What is an error step? You do not describe what you really did. I reconstruct, you increased added errors in $Z_h$, $Z_{DR}$, $K_{DP}$, and $A_h$ simultaneously. How did you control the error covariances? How did you distribute the errors?

- p13, l3, and l 14: No "

- p14, l28: Malte Diederich

- p17, table1: Give citations for the applied relations.

- p17, table3: The exponent of $Z_{DR}$ is in the wrong line.

- p20, figure3c: Average of $\rho$, not STD_ZDR.

- p22, figure5c: (same error)

- figures 6, 7, 10, 11: Most data are ploted in the lower left corner. I propose to use double logarithmic scales or to add an enlarged version additionally to show the data up to 20 mm rainfall.

- figure 9: It should be "range" not "Gate".

- figure 12: Specific attenuation in dB/km, not degrees/km.
* * *

---

## Author Comment (AC2) · 2 Mar 2016

Response to review At first, thank you very much for referee's effort in reviewing our paper even your busy time. We revised the manuscript titled "The possibility of rainfall estimation using R(Z,Z$_{DR}$,K$_{DP}$,A$_{H}$): A case study of heavy rainfall on 25 August 2014 in Korea" that was submitted to Hydrology and Earth System Sciences. The manuscript has been revised as suggested. We would appreciate any feedback on the revisions. Response to review by Anonymous referee 2

Overview

1. The aim of the authors is to find the optimum quantitative precipitation estimation

method for the first dual-pol S-Band radar in Korea. To this end they search for the best method to remove biases in (horizontally polarized) reflectivity measurements ($Z_h$) and in differential reflectivity ($Z_{DR}$). They test mainly R($Z_h$,$Z_{DR}$) and R($Z_h, Z_{DR}$, $K_{DP}$, $A_h$), the later one proves to provide quite stable and reliable estimates of the rain intensity. A basic requirement for a publication is, that an (educated) reader is able to comprehend what was done and that she/he is able to repeat the investigation, based on the information of the publication. This goal is not met by this manuscript. I did not find a clear structure within the paper. It should be stated clearly, that different approaches to determine the biases in $Z_h$ and $Z_{DR}$ will be discussed in advance of describing the first approach.

Author's Response:

Thank you for your comment. We reorganized "2 Data and methodology and 3 Results" of the manuscript to help a reader understand what was done in the paper as follows; 2 Data and methodology; 2.1 Gage, disdrometer and radar data; 2.2 Calculation of polarimetric; 2.3 $Z_{DR}$ data quality improvement; 2.4 Validation, 3 Results; 3.1 Data quality of $Z_{DR}$; 3.1.1 Improvement of $Z_{DR}$ data quality using moving average; 3.1.2 Improvement of $Z_{DR}$ data quality using disdrometer; 3.2 Validations of two rainfall relations; 3.2.1 The performance of rainfall relations with $Z_H$ and $Z_{DR}$ biases obtained by radar, 3.2.2 The performance of the relations using $Z_{DR}$ bias obtained by disdrometer, 3.2.3 The simulation of R(Z,$Z_{DR}$,$K_{DP}$,$A_H$) using generated variables. We also changed the title of the paper as the first reviewer's suggestion "The validation of polarimetric rainfall estimates using R(Z,$Z_{DR}$) and R(Z,$Z_{DR}$,$K_{DP}$,$A_H$) in Korea".

2. The abstract does not describe what is done in this investigation.

Author's Response:

Thank you for your comment. We modified the abstract to describe what was done in this investigation more clearly as follows; "To improve the accuracy of polarimetric rainfall relations with different $Z_{DR}$ bias calculations using 9 gate averaged $Z_{DR}$ and

a disdrometer, three rainfall cases were analysed and some methods were examined. The observed differential reflectivity ($Z_{DR}$) quality check was theoretically investigated using the relation between the standard deviation of differential reflectivity and cross correlation, and the light rain method for $Z_{DR}$ bias was also applied to the rainfall estimation. The best performance for these heavy rainfall case was obtained when the moving average of $Z_{DR}$ over a window size of 9 gates was applied to the rainfall estimation using horizontal reflectivity ($Z_H$) and $Z_{DR}$ and to the calculation of $Z_H$ bias. The differential reflectivity calculated by disdrometer data may be an alternative to the vertical pointing scan for calculating $Z_{DR}$ bias. Comparing the statistical scores between R(Z,$Z_{DR}$) and R(Z,$Z_{DR}$,$K_{DP}$,$A_H$) in this study, R(Z,$Z_{DR}$) had better performance than that of R(Z,$Z_{DR}$,$K_{DP}$,$A_H$). However, R(Z,$Z_{DR}$,$K_{DP}$,$A_H$) is expected to be less sensitive especially to $Z_H$ and $Z_{DR}$ errors in both observations and simulations. Therefore, R(Z,$Z_{DR}$,$K_{DP}$,$A_H$) could be used as a representative rainfall relation in case $Z_{DR}$ bias was not calculated accurately in Korea."

3. The paper contains a large amount of oversight, reducing the confidence in the care the authors took. The scientific innovation within this paper is quite limited. Known QPE approaches are tested and the results gathered during a strong precipitation event are presented. It is questionable if this is innovative enough to justify publication. Because of the limited quality in presentation I would reject the paper in the present form.

Author's Response:

Thank you for your comments. There are many researches on rainfall estimation using polarimetric variables. The researches were focused on R(Z,$Z_{DR}$), R($K_{DP}$), R($K_{DP}$,$Z_{DR}$), R(Z,$Z_{DR}$,$K_{DP}$). Recently, Ryzhkov et al. (2014) firstly found that the specific attenuation gives us more accurate rainfall estimates even for S-band polarimetric radar. After their study, $A_H$ became one important variable for quantitative usage for S-band polarimetric radar. Most studies of rainfall estimation using AH, the performance of R($A_H$) was examined. The combined rainfall relations like R(Z,$Z_{DR}$,$K_{DP}$,$A_H$) was rarely investigated until now. This study was focused on the performance of rainfall

relations such as R(Z,ZDR) and R(Z,$Z_{DR}$,$K_{DP}$,$A_H$) with respect to $Z_H$ and $Z_{DR}$ bias and $Z_{DR}$ data quality using observed data and generated data. We think these results obtained from this would be a possible topic for publication. We also added two more cases in the manuscript to make the paper more confident. The three rainfall events occurred on 23 August 2012, 8 September 2012, and 25 August 2014, which were caused by indirect effect of Typhoon, low pressure accompanied with the front, and low pressure were included and summarized in Table 1. We added related figures and description in the manuscript and also modified some mistakes.

Specific notes

1. p2, l16: "different drop shape", different from what?

Author's Response:

We are sorry for the confusion. We added the descriptions to make it clearly from line 19 to line 21 on page 2 in the revised manuscript as follows; "1) equilibrium shapes defined by Beard and Chuang (1987), 2) oscillating raindrop shape from Bringi et al. (2003) and 3) shapes specified by Brandes et al. (2002)."

2. p2, l28: KMA installed an S-Band polarimetric radar in the fare northwest of Korea. Later in the next, the Bislsan radar was the first polarimetric radar in Korea. Bislsan is in the southeast of Korea. Is this a contradiction or are there at least two polarimetric radars in Korea?

Author's Response:

We are sorry for making reviewer confused. We would like to describe the current status of polarimetric radar network in Korea. KMA (Korea Meteorological Administration) installed one S-band polarimetric radars in 2014 and KMA is replacing 10 single polarization radars into polarimetric radar. The replacement will be done until 2019. MoLIT (Ministry of Land, Infrastructure and Transportation) installed 5 S-band polarimetric radars since 2009 and will install one more radar soon. Ministry of National

Defence (NMD) has plan to replace 6 C-band single polarization radars into polarimetric radar. Anyway, because the description "The KMA installed an S-band polarimetric radar in the far northwest of Korea in 2014" was not important information we removed the sentence.

3. p3, l26: Fig. 1 show the location of all instruments? Where are the rain gages mentioned at the beginning of chapter 2.3?

Author's Response:

Thank you for your comment. We added the location of gages described in Section 2.3 of original manuscript in Figure 1 and added the following sentence line 3 to 5 on page 4 of revised manuscript. "The cross (ID 945, Daebyung site), triangle (ID 926, Jinbook site), and diamond (ID 255, North Changwon site) with plus sign show the location of gages which recorded the maximum daily rainfall accumulation in each rainfall events will be analysed in Sect. 2.4."

4. p4, l2: Radar Bislsan is (according to my digital elevation model) at a height of more than 1 km asl. The disdrometers are quite close to see level. In 82 km from radar the 0.5° beam is 1.1 km above radar height. There is nearly 2 km separation between radar and disdrometer measurements? Are these data comparable? You should at least discuss this problem.

Author's Response:

We agree that there is some limitations of using disdrometer data especially for the convective systems which have much fluctuation of DSD with height. There would be fluctuation of DSD with height in three cases we analyzed. As reviewer's comment, we added the limitations in the Sect. 4. Conclusions of the manuscript as follows; "Using DSD data for the calculation of $Z_{DR}$ bias might give more accurate rainfall estimation with R(Z,$Z_{DR}$), even it is limited to the homogeneous DSD at the layer between radar beam height and ground and not strong wind condition which could degrade the quality

of ZDR calculation from disdrometer."

5. p3, l12: Drop numbers count only in the lower channels leads to an removal of the data. In the next step you remove drop size spectra with drop number counted only in the lower 5 channels? This is done twice.

Author's Response:

Thank you for your comment, As reviewer's comment, we removed the description.

6. p4, l25: The prefactor of D is 0.00057, not 0.5….

Author's Response:

Thank you for your comment. We modified Equation (1) correctly.

7. p5, l24: What is N? N is the number of rain gages(121) or the number of hourly measurements (7 *121)?

Author's Response:

Thank you for your comment. N is the gage number multiplied by analysed hours. We modified the corresponding sentence as follows; "N is the number of radar rainfall ($R_R$) and gage rainfall ($R_G$) pairs, and and are the hourly rainfall amount in each rainfall event from the radar and gage, respectively."

8. p6, l20: There is no Bringi and Chandrasekar (2003). You assumable meant 2001, or did you mean Bringi et al. 2003?

Author's Response:

We are sorry for confusion. Bringi and Chandrasekar (2001) is correct. We modified it.

9. p6, l21: At least in Bringi and Chandrasekar (2001) I did not find this equation. Please give a more precise citation. What is $\rho_{co}$, what is $\rho_{co}(l)$? is $\rho_{co}$= $\rho_{co}(0)$? You call the correlation at different palces, $\rho_{co}$, $\rho_{co}(l)$, $\rho[n]$ and $\rho_{hv}$. Is it all the same thing? So please indicate what meant by which term.

Author's Response:

Thank you for your comment. We unified all terms into hv. We also added the equation number of Bringi and Chandrasekar (2001) in the revised manuscript.

10. p7, l6: With equation 8, an L of over 3 (line 11) is reached by $\rho_{hv}$ > 0.5 and an L of 1.7 (line 14) needs $\rho_{hv}$=.32. Probably the prefactor 10 is wrong in equation 8. Author's Response:

Thank you for your kind comment. We removed prefactor 10 in Eq. (8).

11. p8, l2: Ryzhkov et al. (2005a)

Author's Response: Thank you for your comment. We modified it in the revised manuscript.

12. p8, l19: Why do you have problems to detect the melting layer by a 6 elevation volume scan? There are approaches to determine the melting layer from an individual elevation.

Author's Response:

Thank you for your comment. In summer of Korea, the brightband is usually occurred at the layer between 4 and 5 km. Considering the maximum elevation and range of BSL are 1.6 degree and 150 km, the bright band may be located at the range of around 130 km even we consider the height of BSL. And dry aggregated snowflakes are commonly presented in the first two kilometres above the melting layer. It would be considered that it is very difficult to use dry aggregated snow method. That is why we described that the scan strategy with six elevation angles not to detect the melting layer. For reader's understand, we modified the description as follows in the revised manuscript; "The scan strategy of BSL with six elevation angles (-0.5, 0, 0.5, 0.8,1.2,1.6 degree) is not allowed to use dry aggregated snow method to calculate $Z_{DR}$ bias."

13. p10, l24: Table 3 contains the results from chapter 3.3. The results from chapter

4.1 are not given.

Author's Response:

Thank you for your kind comment. We added Table 5 and Table 6 to summarize the results of Chapter 3.2.2 in the revised manuscript.

14. p11, l5: As far as I got it, you never introduced Ah, although that term is already used in the title. I assume, $A_h$ is the path integrated attenuation for the horizontally polarized wave. This should be measured in dB/km but not in degrees/km.

Author's Response:

Thank you for your comment and we are very sorry for confusion. We modified $A_H$ unit from degrees/km to dB/km and also described the following sentence about the AH in the revised manuscript. And we attached the supplement which includes more detail description about $A_H$ calculation. "$A_H$ was calculated from the radial profile of the attenuated reflectivity and two-way PIA (Path Integrated Attenuation) along the propagation path using observed $Z_H$, differential phase shift from BSL radar. The more detailed description for $A_H$ calculation can be found in You et al. (2015a)."

15. p11, l18: What is an error step? You do not describe what you really did. I reconstruct, you increased added errors in $Z_h$, $Z_{DR}$, $K_{DP}$, and $A_h$ simultaneously. How did you control the error covariances? How did you distribute the errors?

Author's Response:

Thank you for your comment and we are sorry for confusion. The errors of Z, $Z_{DR}$, and $K_{DP}$ ingested to simulated data were distributed 0 to 5 dBZ with interval 0.25 dBZ, 0 to 0.6 dB with interval 0.03 dB, and 0 to 0.2 degree/km with interval 0.01 degree/km, respectively as mentioned in the manuscript. The errors were ingested to the simulation with 21 steps. For example, the errors of $Z_H$, $Z_{DR}$, $K_{DP}$ were ingested to the simulation as much as 0.25 dBZ, 0.03 dB, 0.001 degree/km $in number 1 of x-axis on the Figure 14. At the same way, the errors were increased with interval of each variable as the number of x-$

*axis increases. We did not consider the error covariances in this study because we would like to simulate under the worst situationa*

is relatively weakly affected by errors in each polarimetric variable even though the error co-variances were not considered in the simulation."

16. p13, l3, and l14: No "

Author's Response:

Thank you for your comment. We guess that No means Journal number. According to the Copernicus Publications Reference Types, we do not have to describe the Journal number.

17. p14, l28: Malte Diederich

Author's Response:

Thank you for your comments. We modified it in the revised manuscript.

18. p17, table 1: Give citations for the applied relations.

Author's Response:

Thank you for your comment. We added citations in Table 2 of revised manuscript and also added Figure 3 as the first reviewer's comment.

19. p17, table3: The exponent of $Z_{DR}$ is in the wrong line.

Author's Response:

Thank you for your comment. We modified the Table 3.

20. p20, figure3c: Average of $\rho$, not $STDZ_{DR}$.

Author's Response:

Thank you for your kind comment. We modified the Figure.

21. p22, figure 5c: (same error)

[Figure]

Author's Response:

Thank you for your kind comment. We modified the Figure.

22. figure 6, 10, 11: Most data are plotted in the lower left corner. I propose to use double logarithmic scales or to add an enlarged version additionally to show the data up to 20 mm rainfall.

Author's Response:

Thank you for your comments. We added enlarged version additionally to show the scatters up to 20 mm h$^{-1}$.

23. figur9: It should be "range" not "Gate"

Author's Response:

Thank you for your comment. We modified Figure 9.

24. figure 12: Specific attenuation in dB/km, not degrees/km

Author's Response:

Thank you for your comment. We modified the unit of Specific attenuation.

\*\*\* Thank you very much again for your deep review and it will be of much help for better our manuscript quality.\*\*\*

Please also note the supplement to this comment:
http://www.hydrol-earth-syst-sci-discuss.net/hess-2015-515/hess-2015-515-AC2-supplement.zip